# Efficient Model Updating of a Prefabricated Tall Building by a DNN Method

**DOI:** 10.3390/s24175557

**Published:** 2024-08-28

**Authors:** Chunqing Liu, Fengliang Zhang, Yanchun Ni, Botao Ai, Siyan Zhu, Zezhou Zhao, Shengjie Fu

**Affiliations:** 1School of Civil and Environmental Engineering, Harbin Institute of Technology, Shenzhen 518055, China; chunqingliu98@outlook.com (C.L.); zhangfengliang@hit.edu.cn (F.Z.); 22s054013@stu.hit.edu.cn (Z.Z.); 22s054016@stu.hit.edu.cn (S.F.); 2College of Civil Engineering, Tongji University, Shanghai 200092, China; 3Guangdong Provincial Key Laboratory of Intelligent and Resilient Structures for Civil Engineering, Harbin Institute of Technology, Shenzhen 518055, China

**Keywords:** model updating, DNN, sensors, partition walls

## Abstract

The significance of model updating methods is becoming increasingly evident as the demand for greater precision in numerical models rises. In recent years, with the advancement of deep learning technology, model updating methods based on various deep learning algorithms have begun to emerge. These methods tend to be complicated in terms of methodological architectures and mathematical processes. This paper introduces an innovative model updating approach using a deep learning model: the deep neural network (DNN). This approach diverges from conventional methods by streamlining the process, directly utilizing the results of modal analysis and numerical model simulations as deep learning input, bypassing any additional complex mathematical calculations. Moreover, with a minimalist neural network architecture, a model updating method has been developed that achieves both accuracy and efficiency. This distinctive application of DNN has seldom been applied previously to model updating. Furthermore, this research investigates the impact of prefabricated partition walls on the overall stiffness of buildings, a field that has received limited attention in the previous studies. The main finding was that the deep neural network method achieved a Modal Assurance Criterion (MAC) value exceeding 0.99 for model updating in the minimally disturbed 1st and 2nd order modes when compared to actual measurements. Additionally, it was discovered that prefabricated partitions exhibited a stiffness ratio of about 0.2–0.3 compared to shear walls of the same material and thickness, emphasizing their role in structural behavior.

## 1. Introduction

The prevalence of prefabricated construction in China has increased significantly over the past decade. This growth can be attributed to several factors, including the rapid construction time, eco-friendliness, industrialization, and standardization. China’s slowing population growth and aging demographics are pushing up labor costs, particularly affecting the traditional labor-intensive construction. In areas with high labor costs, prefabricated construction is now cheaper than traditional methods. With cost-effectiveness and sustainability [1,2,3], prefabrication is becoming a dominant trend. Structurally prefabricated components offer rotational connections differing from traditional cast-in-place or welded joints. Seismic performance research highlights the need for robust connection design in prefab construction. Innovations like the new prefabricated self-centering steel frames and modular precast shear wall systems have demonstrated superior dynamic response and energy dissipation in tests and analyses [4,5,6].

A substantial body of research has emerged in recent years on the seismic performance of prefabricated partitions in buildings. Zhai et al. [7] found that prefabricated reinforced concrete (RC) shear walls with different infills, particularly integrated shear walls with RC infilling, exhibit superior shear bearing capacity, stiffness, energy dissipation, and seismic performance. The main focus of research on partition walls is not on mechanical properties. Many studies in this field have chosen to focus on the performance of partitions for daily use such as sound insulation [8,9,10], thermal performance [11,12,13,14], and environmental performance [15,16,17,18]. Moreover, Li et al. [19] noted that non-structural infill walls could increase a building’s overall stiffness by 60%, suggesting a similar effect for partition walls. Despite advances in finite element analysis, significant discrepancies remain between model results and actual measurements, as highlighted by Yang et al. [20], who attributed these differences to the simplification of structural behavior, discretization of continuous systems, and physical parameter errors. Therefore, effective seismic resistance solutions, more accurate methods to reflect dynamic characteristics, and improved monitoring and maintenance of prefabricated structures have become key research focuses.

Since the concept of model updating for finite element models was proposed, the earlier adopted methods have been mainly classified as a deterministic model updating method. This type of method tries to get the calculated structural responses, modal shapes, frequencies, and other parameters as close as possible to the measured data by adjusting the structural parameters, as cited in the literature [21,22]. This method is usually an ill-posed inverse problem in model updating, and the incompleteness of data and the complexity of the structure frequently make the results of this updating method inconsistent and incomplete, as mentioned in paper [23]. 

Subsequently, a model updating method based on the Bayesian formula was proposed. Katafygiotis et al. [24] provide a comprehensive explanation of this structural model updating method, taking the measured data of the structure as inputs to calculate the posterior probability distribution function (PDF) in the Bayesian formula, and the maximum value of the parameterized posterior probability distribution is then taken as the updated parameter. In order to determine the posterior PDF in the Bayesian formula with the parameter to be updated θ, the Markov chain Monte Carlo (MCMC) method was used in the literature [25]. This method can be used regardless of whether the problem is identifiable or not, and it can obtain the most likely value of the updated parameters and quantitatively evaluate the uncertainty of this value. Boulkaibet et al. [26] employed the mixed Markov chain theory to update the structural parameters and provide a new evaluation formula for Markov chain convergence. In order to improve the efficiency of this calculation method, Zhang et al. [27] incorporated the Metropolis–Hastings (MH) algorithm. Yang et al. [20] considered and reduced the impact of white noise on the calculated posterior probability distribution function when using this method for model updating of the coupled plate system. 

In recent years, there has been a rapid increase in the number of publications focusing on deep learning-based methods for model updating, showcasing the potential of deep learning in the domain of model updating. Lee et al. [28] have presented a novel methodology for structural damage detection, which leverages finite element model updating to establish a reference model that encapsulates the target structure’s characteristics. This approach addresses the limitations of traditional simulation-based damage detection by incorporating measured responses and employing DNN to identify the extent and location of structural damage. Utilizing an inverse eigenvalue problem approach and DNN, Gong and Park [29] innovatively updated finite element models with high accuracy, as evidenced by the dynamic updating of a suspension bridge model. 

Employing a multi-fidelity deep neural network (MF-DNN) for surrogate modeling, Torzoni et al. [30] introduced a methodology-enhancing real-time structural health monitoring by effectively locating and quantifying damage through an MCMC-informed probability update, leveraging high- and low-fidelity simulated datasets for comprehensive sensor data mimicry. In summary, deep learning—a swiftly evolving methodology—is increasingly applied for various purposes within the SHM field, such as encompassing seismic response modeling and finite element model updating [31,32,33].

This study introduces a novel method for modal updating, termed Deep Neural Network Model Updating (DNNMU), which is an algorithm that, at its core, utilizes the DNN algorithm, and intuitively and simply uses the input and output of finite element models as training data to achieve its model updating objective. The efficacy of the method is demonstrated through its application in a case study involving a prefabricated dormitory, which is located on the Shenzhen campus of the Harbin Institute of Technology.

A comprehensive ambient vibration test was conducted on this building using four three-axial accelerometers. The Bayesian operational modal analysis method [30,31,32,33,34,35,36] was utilized to identify the building’s modal parameters, which mainly included natural frequency, damping ratio, and modal shape. A finite element model of the building was established for model updating. The analysis results were compared with the calculation results, empirical formulas, and measured results of the design institute. The stiffness of the partition wall was updated, and the approximate ratio of the stiffness provided by the partition wall to the partition wall’s inherent stiffness was determined. The efficacy and accuracy of the DNNMU method in model updating were validated by comparing the updating accuracy with that of another similar research.

## 2. Methodology

Deep Neural Network Model Updating is an innovative model updating method proposed in this work. The fundamental principle of the DNNMU method involves the systematic generation of a series of numerical simulation outcomes. These outcomes result from the iterative computation of parameters that have been randomly modified. The generated data are then utilized to train a deep learning model, which is subsequently employed to predict the parameters in need of adjustment, thereby facilitating the refinement of the model. When contrasted with conventional methodologies, this strategy is characterized by its straightforwardness, intuitiveness, and superior efficiency. It demonstrates commendable adaptability across a diverse array of model updating scenarios. Moreover, the embedded deep learning model within the DNNMU framework is designed to be modular, allowing for substitution to meet specific user requirements without compromising the method’s applicability.

### 2.1. The Compatibility of DNN with Model Updating

In numerical simulation, the process of solving and calculating correctly modeled finite element models that accurately reflect real-world problems can be regarded as a function. It can be treated as a function because it fits the definition: each set of inputs, such as finite element software models and parameters, yields a unique set of solutions, like node displacement, element internal force, and structural mode.

In the context of model updating, the function is applied to the parameters involved in the updating process. Denoting the numerical simulation process as a function S(·), it can be expressed as:(1)$$S\left(E_{1},E_{2},\ldots ,E_{n}\;\right)=f,\;\Phi$$
where f and Φ are the analytical natural frequency and modal shape of the finite element model, respectively. $E_{1},{\;E}_{2},\ldots ,E_{n}$ stands for the elastic modulus of different structures, which are the updating target. Note that, for the structural mass, it is commonly taken as known since it is easier to estimate their values. When other conditions are the same, the ratio of elastic modulus is equal to the ratio of stiffness.

The reverse process of S in Equation (1) could be written as:(2)$$E_{1},E_{2},\ldots ,E_{n}=R(f,\Phi )$$

R is referred to as the inverse process of S rather than the inverse function because it cannot be rigorously demonstrated that every set of inputs to R has a unique set of outputs corresponding to it. Although considering the practical significance of the variables around this equation, R is likely to be a function. 

This work adopts the Modal Assurance Criterion (MAC) as the evaluation standard for vibration modes:(3)$$mac_{ij}=\frac{{\left({\hat{v}}_{i}^{T}v_{j}\right)}^{2}}{\left({\hat{v}}_{i}^{T}{\hat{v}}_{i}\right)\left(v_{j}^{T}v_{j}\right)}$$
where the mode shapes under the ith mode obtained from numerical simulation are denoted as $v_{i}$, while the mode shapes under the ith mode obtained from field tests are denoted as ${\hat{v}}_{i}^{T}$.

Assuming that the selected modal ranks are the top n ranks, then $mac_{ij}$ refers to the element in the ith row and jth column of the square matrix $MAC_{n\times n}$. The mac value between two vectors is employed to gauge the approximation level between the vectors. A mac  value closer to 1 suggests a higher degree of similarity between the vectors (equal to 1 when the vectors are identical, always less than 1 otherwise). In the following discussion, a mac mentioned without a subscript default to being the same rank, that is, $mac_{ii}$.

Recall that the core objective of model updating is to identify an optimal set of parameters for adjustment, thereby aligning the results from finite element model calculations as closely as possible with the real-world test results, f and Φ. As described in the MCMC model updating method in reference [27], a set of parameters is generated through a predefined PDF. These parameters are then input into the model, yielding a set of outputs and an associated output distribution function. This output distribution function is then compared to the actual values to derive the distribution function of another set of inputs. This iterative process continues until a superior set of inputs is identified, which makes the mac value approach 1 as closely as possible.

The DNNMU algorithm introduced in this study deviates from traditional methods, instead opting to leverage DNN to approximate the inverse process R. The approximation process involves the generation of sufficient training sets by the iterative invocation of the function computed by the finite element software. Each sample within these training sets includes features and labels. Here, the label corresponds to the input parameter $E_{1},{\;E}_{2},\ldots ,{\;E}_{n}$ for each computation, and the feature equates to the output f, Φ derived from that computation. Evidently, as the feature corresponds to the output of the function S and the label to the function’s input, the neural network trained using these samples will approximate the inverse process R. Consequently, for each input set (f, Φ), it can infer the parameter $E_{1},{\;E}_{2},\ldots ,{\;E}_{n}$ that was used in the computation beforehand.

### 2.2. Algorithm Architecture

The updating algorithm, depicted in Figure 1, consists of two primary components: dataset generation and neural network training. These components can operate independently of each other.

The first part, dataset generation, involves creating a dataset for neural network training. This dataset is generated by continuously producing random parameters per predefined rules. These parameters are then utilized in numerical modal analysis with finite element software. After each calculation, the parameters, frequencies, and mode shapes involved are stored as a set of samples. In these samples, the parameters serve as labels, while the frequencies and mode shapes are the features. Subsequently, a subset of these samples is later randomly selected during the training phase to act as a validation set.

The second part encompasses the training of the neural network. This phase uses the dataset generated in the first component. While this dataset is designed with DNN characteristics in mind, it is not the only viable option. Other feasible alternatives include a range of deep learning and broader machine learning algorithms. The specifics of the neural network architecture employed in this study will be discussed in detail in the subsequent section. 

Upon conclusion of the second component, a trained DNN neural network is acquired. The outcomes from the modal identification of measured data are also compiled in the form of labels in the samples. These are entered into the neural network to yield the network’s predicted parameters. These predicted parameters are then re-introduced into the finite element model for further modal analysis. The comparison of the output results with the measured data forms the basis for evaluating the updating effect of the neural network using the MAC. 

In summary, the DNNMU algorithm exhibits high versatility. Within the scope of model updating, parameters that need adjustment invariably exist, serving as the labels in the algorithm’s sample set. Moreover, parameters for assessing the updating outcomes also exist, acting as the features within the samples. This study presents a specific instance within this algorithmic framework, with the choice of DNN explained in Section 4.2. It is noted that this framework retains its applicability across diverse model updating scenarios, such as those involving bridges, geotechnical structures, or aircraft wings, and is compatible with various updating algorithms.

### 2.3. Generation and Characteristic Analysis of Training Sets 

The construction of a DNN requires consideration of the network’s purpose and the characteristics of the dataset. The function of the neural network is to fit a complex process from a one-dimensional input to a one-dimensional output. After generating parameters using the method described in Section 4.1, which will be detailed later, each set of parameters is inputted into a finite element model to conduct a modal analysis, producing the following computational results: frequencies and mode shapes. Assuming the number of modes analyzed is denoted as $n_{m}$ and the number of nodes involved in the model updating is $n_{p}$, the calculated frequencies for each modal order are:(4)$$f_{i},\;\;i=1,2,\ldots ,n_{m}$$

The mode shape matrix for the  jth node under the ith mode is:(5)$$\varphi_{ij}=\left[\begin{array}{c} dx\\ dy\\ dz \end{array}\right]\;,\;\;i=1,2,\ldots ,n_{m},\;j=1,2,\ldots ,n_{p}$$

The mode shape matrix for the  jth node under the ith mode is written as a 3 × 1 column vector, and after concatenation end-to-end, two features for training are formed.

In the preprocessing of parameters, we first consider the mode shape Φ. This feature reflects the relative relationships among all nodes under this mode. While its absolute size holds no meaning, its relative ratio—which represents the shape of structural vibration—is significant and thus should remain unchanged. This is achieved by normalization.

In contrast, the frequency f, another feature, has meaning both in its absolute magnitude and in the proportional relationships between different frequencies. Consequently, it is neither subjected to relative scaling nor translation. For this project, the first-order frequency measures approximately 0.7 Hz and the overall frequency value is close to 1. Therefore, we did not process the frequency. However, in other projects, physical quantities with properties like frequency may require absolute scaling, especially if their magnitudes are significantly larger or smaller than 1.

Finally, we considered the output parameters, which represent the material’s stiffness. Both the absolute size and relative ratio of this physical quantity hold practical significance. As such, we performed absolute scaling, setting the scaling factor as the reciprocal of the maximum initial value, $2\times 10^{10}$.

The fitting objective should also be considered. The mode shapes calculated from the finite element model are guaranteed to be smooth curves, while the measured data often have obvious non-smooth points due to errors, as shown in Figure 1 and Figure 2. Assume that when previously generated, the set of all possible parameters is $R_{I}$, and the set of all possible output mode shapes is $R_{O}$. The inverse process that needs to be fitted clearly satisfies:(6)$$R:R_{O}\to R_{I}$$

Suppose the matrix composed of the true values of the parameters to be updated is $E_{n\times 1}^{\prime }$. During the training of the neural network, the function fitting only occurs within the ranges of these two sets, $R_{O}$ and $R_{I}$. However, because the measured data are not smooth, there is still an error between the finite element simulation and the actual situation. Even if the true value $E_{n\times 1}^{\prime }$ of the parameters to be updated belongs to the set $R_{I}$, the measured result *Φ* of the mode shape is most likely not in the set $R_{O}$, but in the interval not covered by the training set.

Since the function f is obviously continuous, the fact that the measured result is not in the interval will not result in the failure of function prediction, but it may cause a large deviation in the updating result after the neural network overfits the function, or even be inconsistent with the reality. Moreover, the finite element model itself has various differences from the actual building. When overfitting is performed, it is equivalent to requiring the finite element to infinitely approximate the true value. Then, the other aspects of the error between the finite element model and the actual building will be increasingly reflected in the updated parameters, causing the parameters to vary greatly. Therefore, when constructing the neural network, it is crucial to avoid overfitting and control the degree of fitting.

## 3. Field Vibration Test for a Prefabricated Building

### 3.1. Target Building and Experimental Equipment

The Dormitory Building named Liyuan No. 6 at the Shenzhen campus of Harbin Institute of Technology (as shown in Figure 3) has a cross shape in the horizontal direction. The external length of the main structure is 32.1 m, and the external width is 29.8 m. The building consists of one underground floor and thirty above-ground floors, with an eave height of 97.80 m (measured from ±0.00). The seismic fortification intensity is 7 degrees, and the designed earthquake group is the first group, with a structural safety level of two. The lateral resistance components of the structure are evenly distributed, and the dimensions in the two main axis directions are similar, which can make the modal shape challenging to identify due to the closely spaced modes. 

In this test, four Fortimus seismometers were used, each unit containing an accelerometer, a data collector, and storage equipment, all from the seismic equipment company Güralp (Güralp Systems Ltd. 3 Midas House, Calleva Park Aldermaston, Reading RG7 8EA United Kingdom). The instrument photo is as shown in Figure 4. According to manufacturer data, all four accelerometers have a sensitivity of $0.112\times 10^{-6}\;m/s$. The default sampling rate for the instruments is set at 200 Hz, significantly higher than the general natural frequency range of 0.1 Hz. Therefore, within the potential natural frequency range of the building, 200 Hz satisfies the Nyquist theorem. As such, the default sampling rate was used, and the instrument time was set to Greenwich Mean Time (GMT).

### 3.2. Field Testing Arrangement

The testing methodology involved using four available instruments to perform a series of tests, aimed at obtaining the modal shapes at 30 different locations in the building. As depicted in Figure 5, the solid dots in (a) represent the measurement points, labeled Sn to denote the nth setup. The abbreviations TM, F, and S represent the terminal, floor, and plane measurement positions, respectively. These measurement points are arranged uniformly in the vertical direction; horizontally, due to the need for GPS synchronization of the instruments, we chose to conduct tests at positions 1S and 2S, and this was done on each floor.

Due to the limited number of instruments, the plan was proposed for 10 setups. One instrument (TM1) was always placed at a fixed point at the top of the building (30F) as a reference channel. The remaining three instruments were alternately arranged at other measurement points in the building to facilitate the use of old wires with GPS synchronization. The specific placement of each measurement point for every setup is summarized in Table 1.

Prior to testing, the four instruments were placed together for a preliminary 20-min test to ensure their timing accuracy. Each subsequent array was then tested for 25 min. Throughout the testing process, the north direction on each instrument consistently pointed in the same direction, perpendicular to the building axis. For data processing, the north and east directions of each instrument were converted to x and y directions to align with the model. This test was completed on 1 September 2023.

### 3.3. Modal Analysis

Due to the large amount of data collected, only two representative examples of the acceleration time history of the building are presented in Figure 6. These data, obtained from field tests, were analyzed using the P-EM Bayesian operational modal analysis method [30] to identify the modal parameters of Liyuan No. 6. Compared with the conventional Bayesian modal analysis method, the P-EM approach offers superior performance in separating closely spaced modes and offers enhanced efficiency. It retains all the benefits of the standard Bayesian method, such as the ability for quantitative analysis of result uncertainty. The results of the modal analysis are presented in Figure 7 and Table 2.

### 3.4. Finite Element Model and Numerical Analysis Results

The numerical simulation portion of this study was performed using ANSYS 2021 due to its superior simulation performance and its flexibility in secondary development. A finite element model of the building was established based on design drawings provided by the design institute. Two models were constructed for this study. Model 1 includes only structural components such as shear walls, beams, and floors. Model 2, on the other hand, includes partition walls as shear wall units in their original positions. Model 1 was used to compare the results of the modal analysis. Subsequent model updating was conducted using Model 2, which was built based on Model 1 by adding the partition wall.

Given that the basement does not participate in the modal analysis of the upper part of the building and could potentially generate local modal interference during the analysis process, it was excluded from the modeling. Similarly, to avoid potential local modal interference, the balcony was simplified as a beam load for modeling. This was achieved by considering the balcony board’s thickness (120 mm) and length (1000 mm), while ignoring the weight of the upper railing. The concrete weight was taken as 24 KN/m^3^, leading to a uniformly distributed load that is applied to the beam as follows:(7)p = 24 × 0.12 × 1 = 2.88 KN/m

Upon completion of the modeling (as shown in Figure 8), the modal analysis was performed using the subspace iteration method in ANSYS. The frequency range was set from 0.01 Hz to 20 Hz, and the damping ratio was 0.05.

The results of modal identification of Model 1 can be found in Table 3, which includes data from field tests and the design institute. Two field test results are presented in the table: one from Liyuan No. 6 and the other from Liyuan No. 7. Both buildings are dormitories and were constructed using nearly identical design drawings.

As Table 3 illustrates, the two field test results are in close agreement, as are the numerical results. However, a significant discrepancy becomes apparent when comparing the field test results to the numerical results. The frequency of bending modes in the field test is nearly twice that of the numerical results, while the frequency of torsional modes in the field test is approximately one quarter that of the numerical results. In the numerical analysis, neither model considered the stiffness provided by the partition walls. Therefore, this study focuses on these walls for model updating. The goal was to investigate the ratio of the stiffness they provide in the building, compared to that provided by shear walls of the same thickness and location.

## 4. Application

### 4.1. Random Generation of Calculation Parameters

As discussed in the comparison and analysis of results in Section 3.4, the noticeable discrepancy between the field test modal identification results of the study’s test subject—Liyuan No. 6—and the numerical results can be attributed to the overlooked stiffness of the partition walls. While there has been considerable research on the inherent stiffness and mechanical properties of precast concrete partition walls, studies on their influence on the stiffness of a finished building have been relatively scarce. This research seeks to identify an appropriate set of stiffness values for these partition walls through model updating, with the goal of adapting the finite element model of Building 6. Simultaneously, we hope to propose a reference value for the actual stiffness of precast concrete partition walls in high-rise buildings, extrapolated from neural network predictions of material stiffness.

Assume that the parameter to be updated is $E_{i},\;(i=1,\;2,\ldots ,\;n)$, and the empirical possible value for each parameter is ${\bar{E}}_{i},\;(i=1,\;2,\ldots ,\;n)$. Assume a class of probability density functions p that have similar property. Choose n of these *p* function, denoted as $p_{i}\left(i=1,\;2,\ldots ,n\right)$. Each time a sample is selected, pick values $r_{i}\left(i=1,\;2,\ldots ,n\right)$ that individually adhere to these n probability density functions. The parameters selected then are:(8)$$M_{n\times 1}=\left[\begin{array}{c} {\bar{m}}_{1}\times r_{1}\\ \begin{array}{c} {\bar{m}}_{2}\times r_{2}\\ \vdots \end{array}\\ {\bar{m}}_{n}\times r_{n} \end{array}\right]=\left[\begin{array}{c} m_{1}\\ \begin{array}{c} m_{2}\\ \vdots \end{array}\\ m_{n} \end{array}\right]$$

In this study of the actual stiffness of partition walls in high-rise building, there was little prior experience to draw on. Therefore, the empirical value was selected as 2/3 of the concrete stiffness of the studied partition wall, and the probability density function was chosen as:(9)$$p=\left\{\begin{array}{c} 1.25,\;\;\;\;x\in \left[0.2,1\right]\\ 0,\;\;\;\;\;\;\;\;\;x\notin \left[0.2,1\right] \end{array}\right.$$

Obviously, when choosing parameters, the selection of the probability density function *p* corresponding to each parameter is crucial. In general, the more certain the property that the parameter represents, the more accurate the empirical estimate, and consequently, the larger the value of *p* around x = 1 will be, and the opposite is true for less certain properties. This selection is due to the uncertainty of the partition wall stiffness, and the stiffness of the partition wall connected by splicing cannot approach its inherent stiffness. Once the parameters to be adjusted are generated, they can be incorporated into the program to modify the finite element model parameters and perform iterative calculations using the finite element software. The calculated results are then saved as a training dataset.

In the modeling process of partition walls, they were treated similarly to shear walls, modeled using shell units, with the elastic modulus of each partition wall serving as the updating parameter. Under consistent circumstances of connections and geometric dimensions, the stiffness contributed by the SHELL unit directly correlates with the elastic modulus. Therefore, the ratio of the stiffness provided by the partition wall unit in the model to that of a shear wall of identical thickness can be determined by dividing the parameter value, derived post-updating, by the original elastic modulus of the material. The subject of this study utilized precast concrete partition walls, of grade C30, with an elastic modulus of 30 GPa. The targets for updating are all partition walls situated beneath beams or between shear walls, serving as dividers. Considering the different positions and opening situations of the partition walls, they are categorized into eight types, along with the glass curtain wall on the first floor, resulting in a total of nine parameters to be updated, as illustrated in Table 4.

In Table 4, it is not difficult to notice that the load-bearing structures of the building were not included in the scope of model updating. The reason for this is that, in general, the deviation of the elastic modulus of the load-bearing structures is about 5%, and in the subject of this paper, this deviation has a significantly smaller impact on the natural frequency of the structure compared to the partition walls. Since the error ranges of the two are clearly different, including the main load-bearing structure in the scope of consideration could likely lead to extremely unreasonable deviations due to the influence of the partition walls.

In the modeling, partition walls were treated as shear walls of equivalent thickness (without openings), with the material properties attributed to the respective wall positions. It is important to note that as this paper categorizes variations such as openings and construction methods, the updated elastic modulus does not truly represent a specific property of material, but rather serves as an equivalency value for the stiffness provided by the materials in the wall area. The updated elastic modulus will be referred to as the equivalent elastic modulus.

Upon incorporation of the partition walls, the Partition wall model in ANSYS of the first and second floors are shown in Figure 9. In the figure, different colors correspond to different materials, with examples including green for curtain walls, dark blue for external partition walls, light blue for windowed external partition walls, and yellow for unopened firewalls.

The selection of the initial elastic modulus for the concrete partition wall, as mentioned above, was set to two-thirds of the elastic modulus of C30 concrete, i.e., 2 × 10^10^ Pa. The elastic modulus of the curtain wall was tentatively set at 5 × 10^9^ Pa due to lack of reference. It is worth noted that an overestimation of the elastic modulus is of little concern during this process. This is because if such a situation arises, the updated values for each elastic modulus will shift away from the pre-determined estimated range set by this research significantly. At that point, the necessary adjustments are made.

### 4.2. DNN Structure Design

Drawing upon previous discussions, the primary aim of the neural network construction in this research is to model the inverse function of calculations performed during modal analysis using finite element software. It is crucial that the network demonstrates a certain level of generalization ability to avoid overfitting. The final structure of the neural network used is shown in Figure 10.

The neural network is designed to take two distinct one-dimensional matrices as input: the frequencies and the mode shapes. Given the unique physical implications of these parameters, they should not be simply concatenated into a larger one-dimensional matrix. For instance, the frequency under a specific mode embodies comprehensive information about the integrated stiffness and shape of the entire building, while an element in the mode shape merely represents the relative displacement of a specific node in each direction under the corresponding mode. Therefore, a conventional deep neural network architecture is used to process the mode shapes, while the frequencies are processed in a shallower neural network that merges with the former in the final layer. Notably, in contrast to the network that processes mode shapes, the network designated for frequency processing does not employ dropout on its neurons. 

Since all nine DNN output parameters represent the modulus of elasticity, the Mean Square Error (MSE) function serves as the loss function. Considering the real-world implications of the partition’s modulus of elasticity, a coefficient of variation is incorporated as a penalty term. Notably, the first of the nine modified partitions is merely a curtain wall positioned on a single floor, and hence is excluded from the calculation of the coefficient of variation. The loss function is:(10)$$loss=MSEloss\left(target,predict\right)*(1+c.o.v(\hat{predict}))$$
where target and predict respectively denote the label and the DNN output, and $\hat{predict}$ signifies the parameters from the second to the ninth of the DNN output.

### 4.3. DNN Generalization Ability Test

As previously mentioned, the generalization capability of the DNN is crucial for the research. The probability density function used to generate parameters is defined in Equation (9).

With this probability density function, 5000 samples were generated. The selection of this number was based on papers [20,25] using the MCMC method, which also requires the comprehensiveness of samples. Sample sets of 2500, 5000, and 7500 were generated, and the results showed a slight improvement in accuracy when the quantity increased to 5000, with almost no change beyond that.

A broader probability density function was adopted to generate another 5000 samples:(11)$$\dot{p}=\left\{\begin{array}{c} 1/1.4,\;x\in \left[0.1,1.5\right]\\ 0,\;\;x\notin \left[0.1,1.5\right] \end{array}\right.$$

These samples were used as a validation set in the tests. At this point, if the computed loss value significantly increases after a certain number of epochs, it indicates that the neural network is overfitting in the region covered by the original probability density. The changes in loss after 600 epochs are displayed in Figure 11a,b. These figures use samples from the sample set as the validation set and incorporate a training set characterized by a wide parameter range.

The figure illustrates that within 600 epochs, the neural network progressively improves its generalization ability for samples beyond the training parameter range. This improvement is evidenced by a steady decrease in MSE loss throughout the training, suggesting no overfitting within the finite element solution’s scope. However, predictably, the equivalent elastic modulus of several partition walls exhibits increasing variability as training progresses. As previously analyzed, this variation stems from the neural network overly fitting the inverse process of the finite element calculation, leading to a shortfall in the generalization capacity for the inputted measured modes. Based on these observations, an epoch iteration count of 200 was deemed appropriate for this case.

### 4.4. Model Updating of a Prefabricated Building

Upon finalizing the model using the DNNMU method, the predicted parameters were generated, specifically the elastic modulus. These parameters were then incorporated into the finite element model. Subsequently, a numerical modal analysis was conducted on this updated model to yield the modal parameters. The resulting values were compared with measured results. Table 5 illustrates the comparison between the updated calculated frequencies and the measured frequencies, along with the MAC matrix. Figure 12 presents the comparison of mode shapes.

The first observation from the results is that the discrepancy between the sixth order modal frequency prediction of this model updating method and the actual measured data are within a 5% margin. Subsequently, the updated modal shapes results are examined. As depicted in Figure 12, the MAC values of the updated first-order bending mode for the building along the x and y axes are 0.993 and 0.997, respectively, while the first-order torsional mode has a MAC value of 0.96. Clearly, the second order updating of the bending and torsional modes are not as effective as the first order, with the bending mode at approximately 0.95 and the torsional mode near 0.9. The rationale for this is evident: the mode shapes derived from the finite element model are invariably smooth curves, whereas the modal analysis results of the study’s subject, Liyuan No. 6, obviously show significant noise interference. This issue is especially pronounced in higher-order modes, resulting in significant non-smooth points in the measured mode shapes. Consequently, regardless of the outstanding performance of the updating algorithm, the MAC value will inevitably be limited under these conditions and cannot closely approach 1. Compared to the article [37] published in 2019 using the MCMC method, the model correction results of that article also achieve a relative frequency error of less than 5%. The MAC values for the first-order modes (corresponding to Mode 01–Mode 03 in this paper) are generally greater than 0.93, with the highest being 0.989. The MAC values for the second-order modes (Mode 04–Mode 06) are around 0.9. In comparison, our method shows similar performance on the torsional modes, specifically Mode 03 and Mode 06, but demonstrates significant advantages in the remaining modes.

When examining the MAC values over six modal shapes, it is evident that less smooth actual test modal shapes correspond to lower MAC values. The updated modal shapes strike a balance, accommodating the noise of the measured modal shapes, with their overall form reflecting the measured modal shapes after noise removal. Consequently, the author claims that the DNNMU model updating method’s efficacy is most prominent in the two smoothest measured modal shapes in this dataset—BX1 and BY1—where the two modal shapes of the updated finite element model align almost perfectly with the measured results. In comparison, articles [25,26,27], which also conduct model updating on buildings, generally report a MAC value reaching 0.95 in the model updating results, with the torsional mode having a lower MAC value. However, this method demonstrates superior accuracy in the updating results.

To verify the robustness, 20 groups of 2500 samples were randomly selected from a total of 7500 samples and used to perform model updating. The average coefficient of variation for the nine parameters to be updated (elastic modulus) was 9.24%, and the average coefficient of variation for the modal frequencies of the corrected model was 2.86%. Considering the inevitable differences in the elastic modulus for each category, which arise because this paper classifies the partitions within a 30-story building into only nine categories, the coefficient of variation result is not considered high. Additionally, since the objects of adjustment are partition walls, the variance in the degree to which they are fixed to the main structure also affects the updated elastic modulus result.

In terms of efficiency, compared to traditional MCMC methods or any iteration-based methods, the greatest advantage of the method proposed in this article lies in its efficiency. The most time-consuming step in model updating is the calculation of the numerical model, as it often requires tens of thousands of calculations on the model. As shown in Figure 1, part I of the DNNMU method requires thousands of calculations on each parameter probability density function to generate and store a dataset. This process only needs to be carried out once, rather than having to perform thousands of numerical simulations in each iteration as traditional MCMC methods do. When part II conducts the training of the neural network, thanks to the simple structure of the neural network itself, the training can be completed within minutes even just using a CPU. For example, if a finite element software simulation takes 20 s per run, an MCMC method using 2000 sample points and iterating ten times would need approximately five and a half hours. Although our method requires 5000 finite element simulations, it only needs half the time of the former. Once a sufficient number of sample points are obtained, each model updating with the DNNMU method only requires training, which takes less than 3 min due to the simplicity of the DNN architecture used, thus making it highly efficient. Compared to article [38], which uses an ensemble learning decision tree, the training and model adjustment time of that research on a simple laboratory structure is 70 s, which can be said to be nearly equivalent to the efficiency of proposed method.

The predicted elastic modulus of each partition walls is shown in Table 6. Based on the model updating outcomes, the stiffness offered by the prefab partition within the building approximates 25–45% of that provided by the shear wall of equivalent material and thickness. Incorporating these partitions results in nearly doubling the first two modal frequencies of the complete building model. This underscores the significant influence of the stiffness provided by the partitions on the dynamic properties of the structure.

## 5. Conclusions

This paper presents the DNNMU method, which uses environmental vibration test data and modal analysis of a prefabricated building to improve model updating. By analyzing the first six modal parameters and comparing them with finite element models and design calculations, the study confirms the critical influence of partition walls on building dynamics.

The main finding was that the DMU achieves modal frequency deviations within 5% and mac values predominantly over 0.9; in particular, for the first two modes with minimal noise interference, the mac values achieved were 0.993 and 0.997.

The originality of this paper lies in the distinctive application of deep neural networks (DNN) to the field of structural model updating, resulting in a streamlined and efficient methodology that requires very few hyperparameters to be adjusted, making it highly adaptable. This approach requires only a single large-scale numerical simulation to generate the dataset, followed by a relatively simple DNN architecture to implement the updates.

In addition, the variance of 0.399 Hz and 0.702 Hz in natural frequency between the building modeled and the actual structure in the study was higher that is commonly found. This study found that prefabricated partitions have a stiffness ratio of about 0.2–0.3 compared to shear walls of the same material and thickness, emphasizing their role in structural behavior.

A drawback of this method is its relative lack of interpretability compared to MCMC or with machine learning methods more broadly. While this research focuses on a single structure, the results lay the groundwork for future studies on the DNNMU method’s application across diverse buildings and partition types, and to explore the utility of different deep learning models for model updating. This opens up prospects for advancing building design and performance. Moreover, the DNN network architecture utilized in this study invites further exploration to identify more suitable, effective, and accurate deep learning models for model updating.

The DNNMU has the potential to be applied to other structures. If the evaluation metrics for the model updating results are still the modal frequencies and mode shapes (MAC matrix) as in this paper, then the DNN architecture in the method does not need any major modifications; simply replacing the dataset will be sufficient. If the evaluation metrics change, it may be necessary to adjust the DNN structure to accommodate indicators with different characteristics. Furthermore, replacing the DNN in the method with another deep learning model is also feasible. The new model should have the capability to distinguish between different indicators and the ability to prevent overfitting.

## Figures and Tables

**Figure 1 sensors-24-05557-f001:**
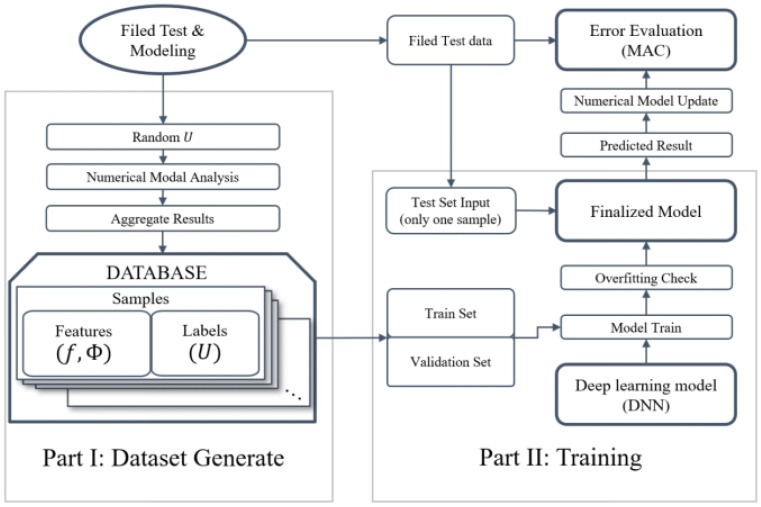
DNNMU algorithm architecture.

**Figure 2 sensors-24-05557-f002:**
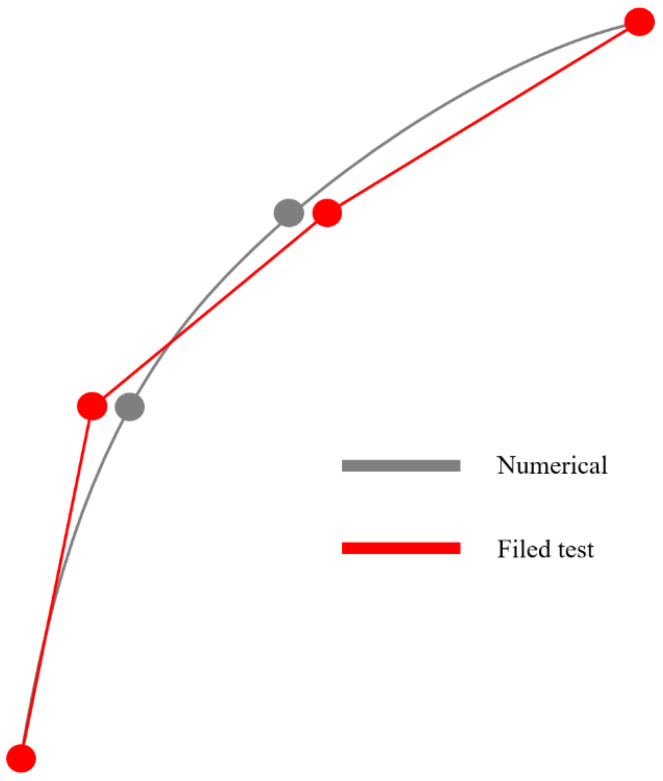
The difference between measured and numerical simulation modal shape.

**Figure 3 sensors-24-05557-f003:**
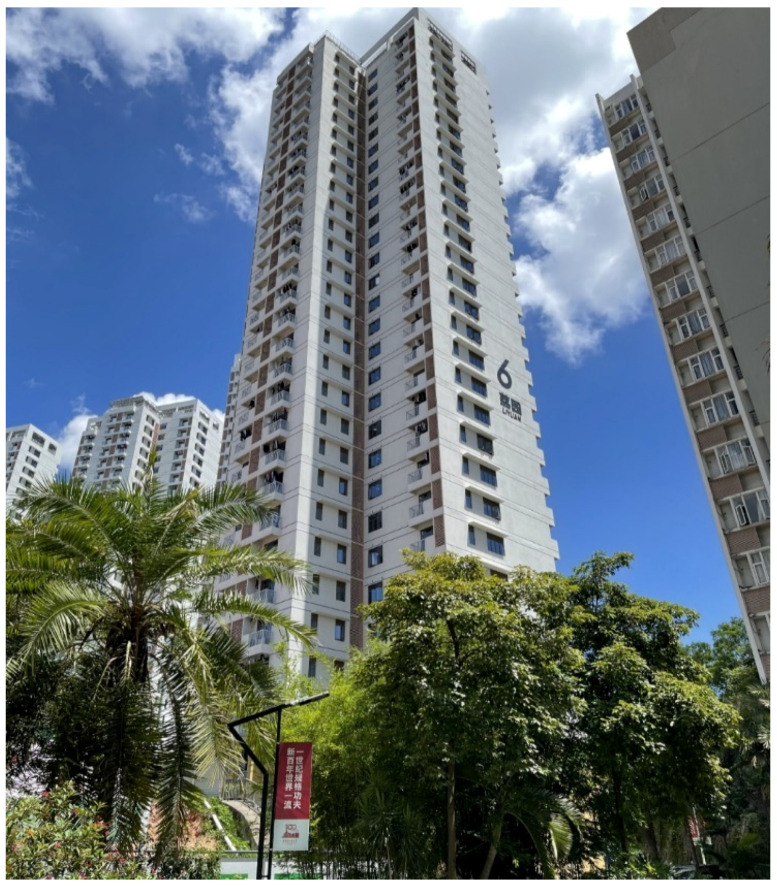
Target building. (The Chinese characters on the building are the building name “Liyuan”).

**Figure 4 sensors-24-05557-f004:**
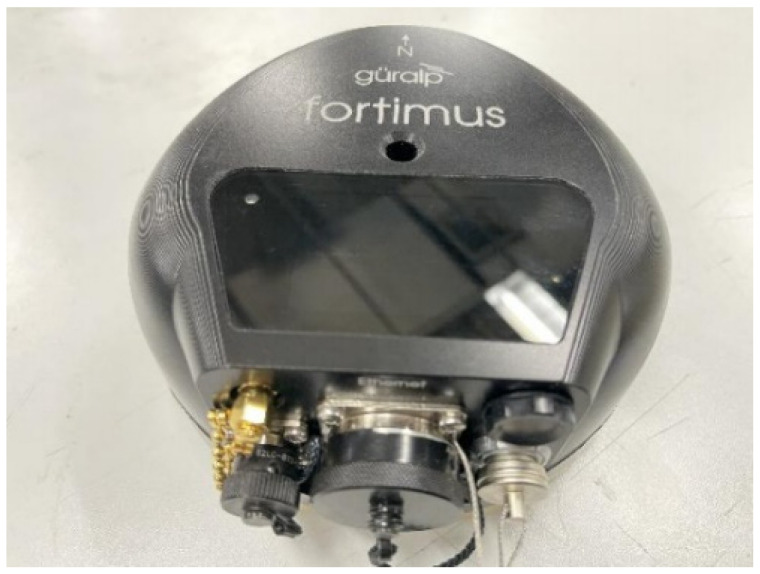
Photo of Fortimus seismometers.

**Figure 5 sensors-24-05557-f005:**
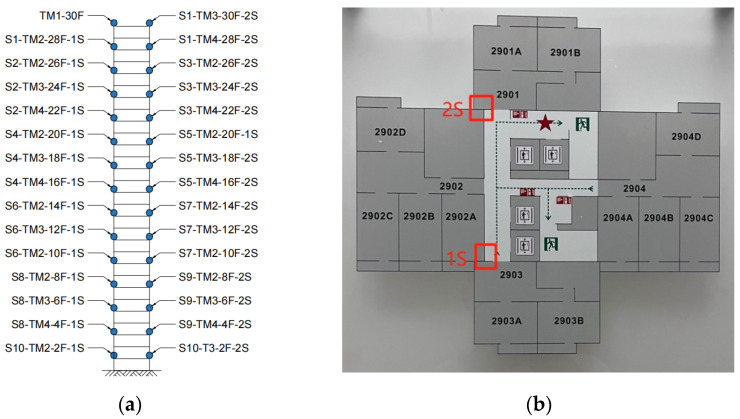
Test point locations. (**a**) Vertical measurement point position; (**b**) Vertical measurement point position.

**Figure 6 sensors-24-05557-f006:**
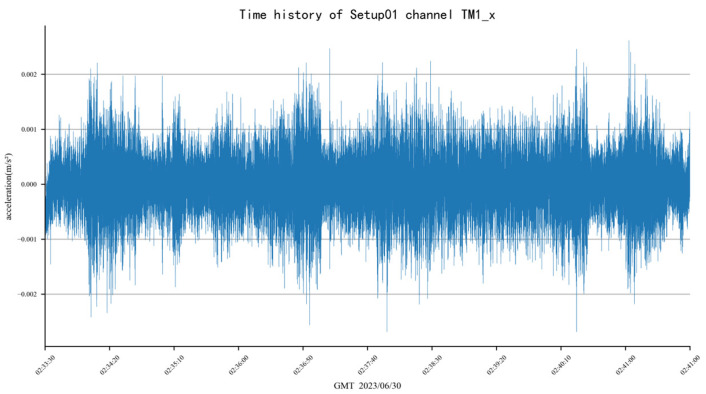
Examples of acceleration time history data.

**Figure 7 sensors-24-05557-f007:**
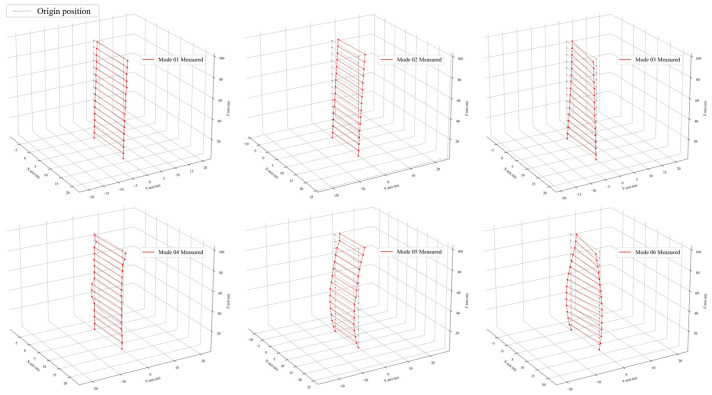
Modal shape results, Liyuan No. 6 dormitory.

**Figure 8 sensors-24-05557-f008:**
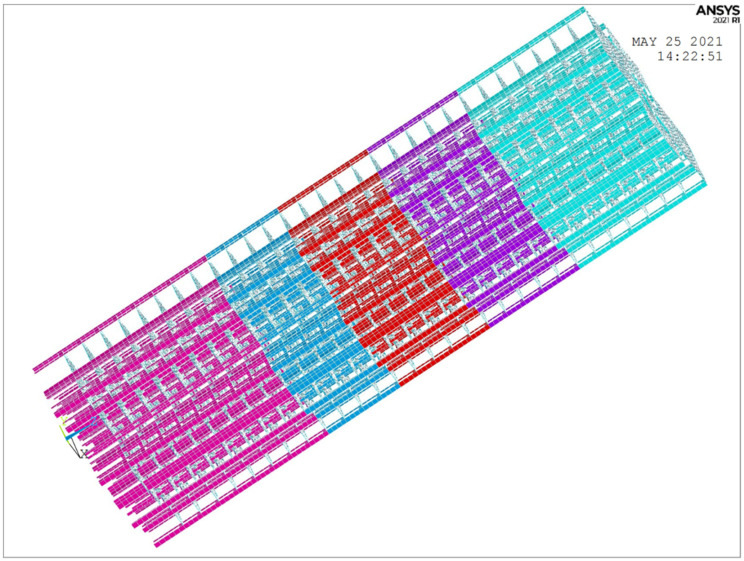
Finite element model without partition walls, Liyuan No. 6 dormitory.

**Figure 9 sensors-24-05557-f009:**
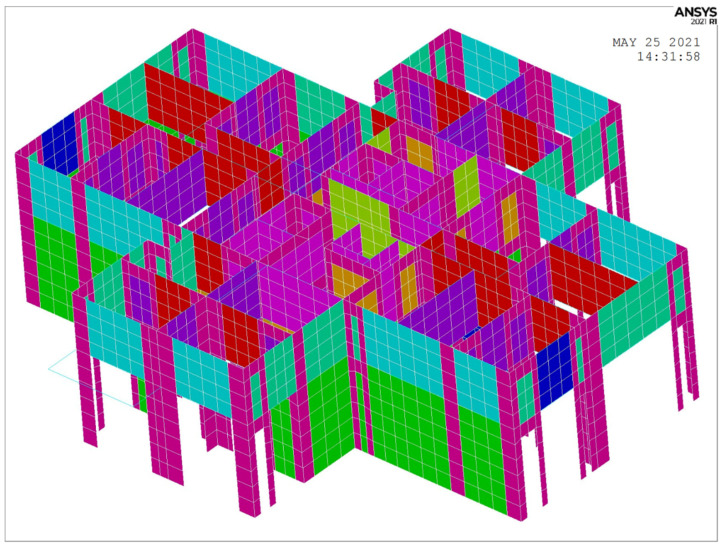
Partition wall model in ANSYS of the first and second floors.

**Figure 10 sensors-24-05557-f010:**
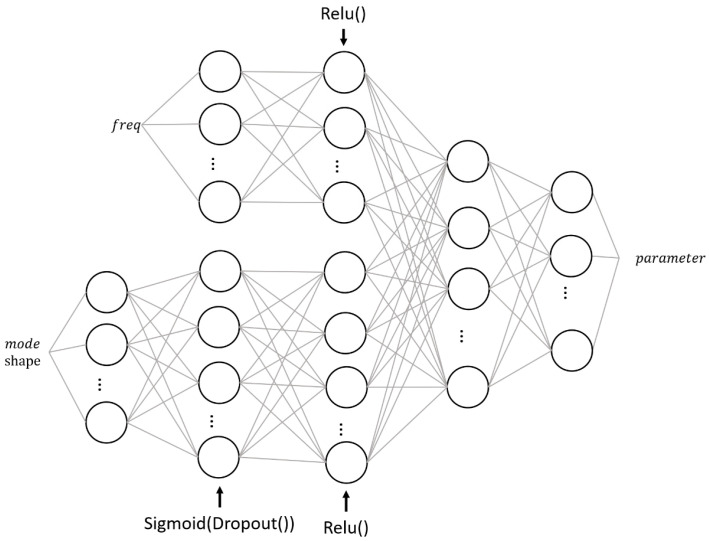
DNN structure.

**Figure 11 sensors-24-05557-f011:**
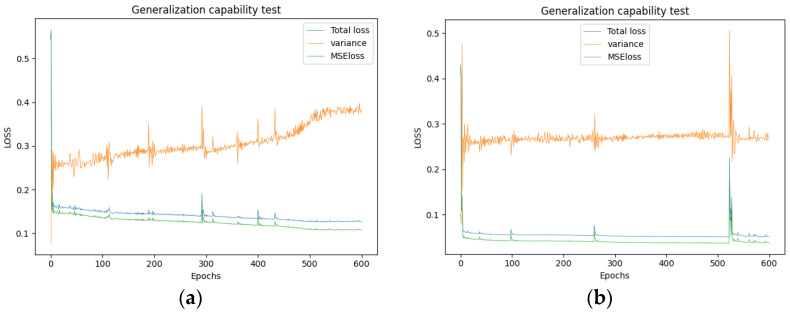
The loss curve of the DNN under the validation set of the same range (**a**) and a broader range (**b**).

**Figure 12 sensors-24-05557-f012:**
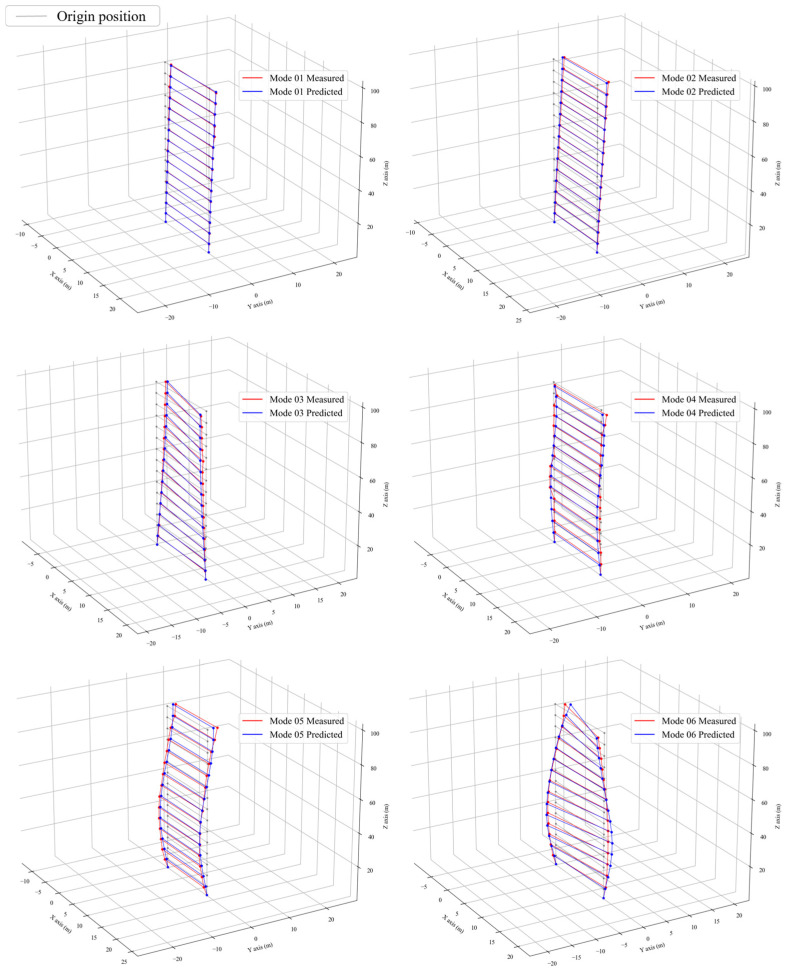
Modal shape comparison.

**Table 1 sensors-24-05557-t001:** Test plan.

SETUP	Measurement Locations			
	TM1	TM2	TM3	TM4
1	30F-1S	28F-1S	30F-2S	28F-2S
2	30F-1S	26F-1S	24F-1S	22F-1S
3	30F-1S	26F-2S	24F-2S	22F-2S
4	30F-1S	20F-1S	18F-1S	16F-1S
5	30F-1S	20F-2S	18F-2S	16F-2S
6	30F-1S	14F-1S	12F-1S	10F-1S
7	30F-1S	14F-2S	12F-2S	10F-2S
8	30F-1S	8F-1S	6F-1S	4F-1S
9	30F-1S	8F-2S	6F-2S	4F-2S
10	30F-1S	2F-1S	2F-2S	

**Table 2 sensors-24-05557-t002:** Modal identification results, Liyuan No. 6 dormitory.

Mode	Frequency		Damping Ratio		Prediction Error PSD		Modal Force PSD	
	MPV [Hz]	c.o.v. [%]	MPV [‰]	c.o.v. [%]	MPV [(μg)²/Hz]	c.o.v. [%]	MPV [(μg)²/Hz]	c.o.v. [%]
1	0.702	0.17	5.622	0.21	59.956	76.20	81.364	28.52
2	0.769	0.15	6.320	0.30	59.956	76.20	83.162	23.59
3	1.430	0.35	6.468	0.29	32.186	67.60	39.935	20.68
4	2.909	0.24	15.003	0.09	92.700	47.04	34.162	32.21
5	3.002	0.28	12.471	0.06	121.079	38.32	40.833	25.36
6	4.282	0.45	14.777	0.33	52.568	38.25	20.771	42.13

**Table 3 sensors-24-05557-t003:** Modal identification results of Liyuan No. 6.

Mode	Frequency [Hz]				
	Filed Test	Numerical	Numerical by Design Institute	Filed Test of Liyuan No. 7	Characteristics
1	0.702	0.399	0.375	0.752	BX1
2	0.769	0.435	0.393	0.822	BY1
3	1.430	0.477	0.478	1.483	T1
4	2.909	1.812	1.727	2.994	BX2
5	3.002	1.883	1.764	3.147	BY2
6	4.282	2.006	1.985	4.392	T2

**Table 4 sensors-24-05557-t004:** Properties of partition walls to be updated.

Wall Type	Initial Elastic Modulus [Pa]	Poisson’s Ratio	Density (Kg/m³)
Curtain Wall	5.00 × 10^9^	0.250	1000
Fire Wall without Openings	2.00 × 10^10^	0.167	2420
Elevator Partition Wall	2.00 × 10^10^	0.167	2420
Fire Wall with Openings	2.00 × 10^10^	0.167	2420
External Partition Wall	2.00 × 10^10^	0.167	2420
Windowed External Partition Wall	2.00 × 10^10^	0.167	2420
Balcony External Partition Wall	2.00 × 10^10^	0.167	2420
Internal Partition Wall	2.00 × 10^10^	0.167	2420
Internal Partition Wall with Door	2.00 × 10^10^	0.167	2420

**Table 5 sensors-24-05557-t005:** Results comparison.

	Mode 01	Mode 02	Mode 03	Mode 04	Mode 05	Mode 06
Measured f	0.7029	0.7672	1.4180	2.9080	3.0071	4.2914
Updated f	0.7097	0.7619	1.3842	2.9616	3.0840	4.1090
c.o.v	0.97%	0.68%	2.38%	1.84%	2.56%	4.25%
Characteristic	BX1	BY1	T1	BX2	BY2	T2
mac	0.9932	0.0046	0.6019	0.0439	0.0220	0.1062
	0.0094	0.9973	0.0766	0.0284	0.0003	0.0201
	0.3986	0.0052	0.9619	0.0018	0.0216	0.0738
	0.0770	0.0058	0.0396	0.9587	0.3529	0.5800
	0.0095	0.0768	0.0020	0.3658	0.9709	0.1527
	0.1605	0.0454	0.1274	0.2655	0.0011	0.8921

**Table 6 sensors-24-05557-t006:** Updated elastic modulus result.

Wall Type	Initial Elastic Modulus [Pa]	Initial Elastic Modulus [Pa]	Ratio (%)
Curtain Wall	5.00 × 10^9^	2.72 × 10^9^	54.40
Fire Wall without Openings	2.00 × 10^10^	9.06 × 10^9^	45.30
Elevator Partition Wall	2.00 × 10^10^	6.94 × 10^9^	34.70
Fire Wall with Openings	2.00 × 10^10^	8.25 × 10^9^	41.25
External Partition Wall	2.00 × 10^10^	8.71 × 10^9^	43.55
Windowed External Partition Wall	2.00 × 10^10^	5.46 × 10^9^	27.30
Balcony External Partition Wall	2.00 × 10^10^	5.78 × 10^9^	28.90
Internal Partition Wall	2.00 × 10^10^	7.33 × 10^9^	36.65
Internal Partition Wall with Door	2.00 × 10^10^	6.07 × 10^9^	30.35

## Data Availability

Data will be made available on request.

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
