# Peer review of "Efficient Model Updating of a Prefabricated Tall Building by a DNN Method"

_sensors, 2024, doi:10.3390/s24175557_

Round 1

Reviewer 1 Report

Comments and Suggestions for Authors

This study proposes a novel model updating method based on deep learning techniques. Different from the complex architectures of other methods, it directly uses the results of modal analysis and numerical model simulation as deep learning inputs, which avoids any additional complex mathematical computations. Consequently, the final algorithm has a minimalist neural network architecture while ensuring accuracy and efficiency. In the application, the method has been implemented in real structures with prefabricated partitions, allowing a high degree of correspondence between the real structure and the updated model. In addition, prior to acceptance, the reviewer suggested some revisions to address a number of issues and clarify certain points.

1. In the last paragraph of section 2.1, it is mentioned that ‘the generation of sufficient training sets’, thus how do you determine whether it is sufficient?

2. Is section 2.2, is the novel algorithm proposed by the authors? If that, please provide more clear statements, and if not, please provide references.

3. For neural network-based methods, there is a crucial issue about the extend of fitting and the final accuracy. For this problem, how is it considered in this paper?

4.In the field test, the measurement plan seems to be complicated, so how were the locations of the measurements in Fig. 5 determined?

5. In section 3.4, why are two models needed for different functions?

6. In DNN structure design, does it imply that the DNN structure needs to be redesigned in different cases? How to obtain a better?

Comments on the Quality of English Language

no

Reviewer 2 Report

Comments and Suggestions for Authors

This paper presents an innovative model updating approach using a fundamental deep learning model—the deep neural network, which diverges from conventional methods by streamlining the process, directly utilizing the results of modal analysis and numerical model simulations as deep learning input, bypassing any additional complex mathematical calculations. Moreover, with a minimalist neural network architecture, a model updating method  has been developed that achieves both accuracy and efficiency. Furthermore, this research investigates the impact of prefabricated partition walls on the overall stiffness of buildings. The paper has significant science soundness and much intrests to readers. I recommend the paper to be acceped after minor revision. 

The comments are as follows:

1. Line 156: th mode should be ith mode.

2. Line 169: "Error! Reference source not found.", the reference is not appropriately cited.

3. Line 227: The same problem as seen in Line 169.

4. The figures such as Fig. 6, 7, 8, 11, 12 should adopt high pixel images, increasing its readability for readers. 

5. The proposed method for model updating should be comapred with the conventional method to illustrate its effectiveness and efficiency.   6. Line 353: Why set the damping ratio to be 0.01?   7. Please have a more detailed comment on the distinctive advantage of the proposed DNN method compared with other machine learing method such as the more convenient ensemble method.   8. The paper just give a case study of the model updating method. How to apply the method to other concrete or steel structures? Please comment.
